# Onychomycosis Endonyx: A Systematic Review

**DOI:** 10.3390/diseases13040110

**Published:** 2025-04-07

**Authors:** Ernesto De-La-Rosa-Garibay, Roberto Arenas, Martha Y. Herrera-Castro, Alicia Valdez-Gaona, Gabriela Moreno-Coutiño, Paola Berenice Zarate-Segura, Fernando Bastida-González, Erick Martínez-Herrera, Rodolfo Pinto-Almazán

**Affiliations:** 1Mycology Section, Hospital General Dr. Manuel Gea González, Mexico City 14080, Mexico; ernestodelarosa105@comunidad.unam.mx (E.D.-L.-R.-G.); rarenas98@hotmail.com (R.A.); marthayareli.herrera@upaep.edu.mx (M.Y.H.-C.); mc19vxga7157@facmed.unam.mx (A.V.-G.); gmorenocoutino@gmail.com (G.M.-C.); 2Internal Medicine, Hospital General de Zona No. 27 “Dr. Alfredo Badallo García”, Intituto Mexicano del Seguro Social, Lerdo 311, Tlatelolco, Cuauhtémoc, Mexico City 06900, Mexico; 3Facultad de Medicina, Universidad Nacional Autónoma de México, Escolar 411A, Copilco Universidad, Coyoacán, Mexico City 04360, Mexico; 4Sección de Estudios de Posgrado e Investigación, Escuela Superior de Medicina, Instituto Politécnico Nacional, Plan de San Luis y Díaz Mirón, Mexico City 11340, Mexico; pbzars@yahoo.com; 5Laboratorio de Biología Molecular, Laboratorio Estatal de Salud Pública del Estado de México, Mexico City 50130, Mexico; mijomeil@hotmail.com; 6Fundación Vithas, Grupo Hospitalario Vithas, 28043 Madrid, Spain; 7Efficiency, Quality, and Costs in Health Services Research Group (EFISALUD), Galicia Sur Health Research Institute (IISGS), Servizo Galego de Saúde-Universidade de Vigo (UVIGO), 36213 Vigo, Spain

**Keywords:** onychomycosis classification, endonyx onychomycosis, *Trichophyton soudanense*, *Trichophyton rubrum*, *Trichophyton violaceum*

## Abstract

Background/Objectives: Endonyx onychomycosis is a chronic infection of the nail plate that presents as milky white discoloration, without hyperkeratosis or onycholysis, and was originally described as being caused by *T. soudanense* and *T. violaceum*. Methods: In the present review, the definitions used in the different articles and the clinical characteristics of patients diagnosed with this onychomycosis variant were analyzed through a systematic review of the reported cases in the literature using the terms “endonyx” AND “onychomycosis” in PUBMED, SciELO, SCOPUS, Web of Science, and Google Academics databases. Results: We found 28 articles with a total of 175 patients diagnosed with endonyx onychomycosis, mainly reported in Asia and Africa. Nine papers presented detailed descriptions. From these, a total of 15 immunocompetent patients were registered, and the etiological agents reported were *Trichophyton soudanense*, *T. rubrum*, *T. violaceum*, *T. tonsurans*, and *Fusarium* spp. After analyzing the definitions employed in the remaining articles, only 47.3% cited or described a concept consistent with the original description. The other 47.3% of the studies lacked a traceable criterion for the diagnosis of these cases. Moreover, most studies analyzed their data at a global level, describing little information to provide specific insights into the endonyx variant. Conclusions: Emphasis on clinical description and histopathological analysis is essential to confirm the role of less commonly reported fungi, and more accurate diagnostic and treatment protocols for this condition are mandatory.

## 1. Introduction

Onychomycosis, the most common nail disease worldwide, is a chronic infection of the nail plate with a wide range of clinical features that can affect finger and/or toenails. It is caused by fungi, most frequently dermatophytes, but it can also be due to yeasts (mainly *Candida* spp.) and Non-Dermatophyte Molds (*Scopulariopsis brevicaulis*, *Fusarium* spp., and *Aspergillus* spp.) [1,2,3].

Clinical classification is an important tool in the medical approach, as the treatment is based on clinical features, and sometimes on the etiological agent involved [2].

Hay et al., in 2011, classified onychomycosis as: distal and lateral subungual onychomycosis (DLSO), superficial white onychomycosis (SWO), proximal subungual white onychomycosis (PSWO), endonyx (EO), mixed pattern onychomycosis (MPO), total dystrophic onychomycosis (TDO), and secondary onychomycosis [4].

The endonyx type was formerly described by Tosti et al. and included in the new classification by Baran et al. in the late nineties [5,6]. The original report was based on the description of three patients with diffuse white milky discoloration of the nail plate, no evidence of hyperkeratosis or onycholysis, and histopathological findings did not show inflammatory changes in the nail bed. Initially, the cases were associated with *Trichophyton soudanense* and with other previously reported cases of nail infection by *Trichophyton violaceum*, with similar histopathological features [7].

It is mandatory to point out the differences between the different forms of onychomycosis. In endonyx, there is an absence of subungual hyperkeratosis; whereas this characteristic is always present in distal and lateral subungual onychomycosis, with the presence of tinea pedis in most cases. Whereas in both forms of leukonychia (WSO, PSWO), white nails are characteristic, and the second form is more frequent in immunocompromised patients [6].

After 25 years of the initial report, we consider that a systematic review of the topic is essential to integrate recent findings and update existing knowledge. Below, we will review the current evidence on endonyx onychomycosis focusing on clinical manifestations, geographic distribution, and the research methodologies used to study this pathology.

## 2. Materials and Methods

### 2.1. Identification

The review was conducted based on Preferred Reporting Items for Systematic Reviews and Meta-Analyses (PRISMA) [8]. An advanced search was performed on November 2024, in five databases: Medical Literature Analysis and Retrieval System Online (MEDLINE/PUBMED), Scientific Electronic Library Online (SciELO), SCOPUS, Web of Science, and Google Scholar, using the terms “endonyx” AND “onychomycosis”, looking for patients diagnosed with endonyx onychomycosis, confirmed by mycological studies, in case reports, case series, clinical trials, and observational studies, from 1 October 1998 to 31 November 2024, published in English and Spanish, with no filters or limits used. We found a total of 891 articles.

### 2.2. Screening and Selection Standards

In order to ensure data accuracy, we eliminated the duplicate articles between the different databases, which numbered 141. We also excluded 25 papers where we could not find the abstract or full text. Exclusion criteria included book chapters, reviews and systematic reviews, or clinical diagnoses not confirmed by a mycological study. Titles, abstracts, and full texts of each potential article were evaluated by three independent reviewers (De-La-Rosa-Garibay, Herrera-Castro, and Valdez-Gaona), and details regarding study inclusion were resolved by a consensus of three reviewers (Arenas, Martínez-Herrera, and Pinto-Almazán).

### 2.3. Data Acquisition

At the end of this process, 34 articles were assessed for eligibility. Four papers could not be retrieved, and three articles were excluded, one because it only presented a clinical picture without further data [9], and the other two [10,11] had the same population as other articles [12,13]. Additionally, 2 references were identified in citation searching; one was not retrieved because we could not locate the publication [14]. Finally, 28 articles were included (Figure 1). Data were collected on the following characteristics: total number of diagnosed cases, geographical location, demographic characteristics (sex and age), anatomical location (fingers or toes and number of affected nails), clinical findings, direct examination results (when available), histopathological findings (in cases with histopathological studies), identified etiological agents (when it could be determined), patients’ immune status, and the presence of associated tinea infections. Data extraction was performed by three reviewers (De-La-Rosa-Garibay, Herrera-Castro, and Valdez-Gaona) using the software Microsoft Excel, Microsoft Office 2021.

### 2.4. Quality Assessment

To assess the risk of measurement bias in the diagnosis of endonyx onychomycosis, we systematically analyzed the definitions utilized in the articles included in the review. Each definition was then compared to the characteristics originally described by Tosti et al. [5]. When a detailed description of the case was available, these were documented and compared in Table 1. In contrast, when the percentage of the reported cases was the only data obtainable, the total number of cases was calculated (based on the total number of patients included in the study), and then reported in Table 2. Additionally, the precise definition was identified either within the text or cited bibliography where they explicitly mention the endonyx concept. This methodology allows us to determine the consistency and accuracy of the terms employed. To assess the quality of this study, the PRISMA 2020 checklist was used [8]. No attempts were made to contact the authors of the studies that presented limited information on the cases. This decision was due to time and resource limitations for the review, as well as the low likelihood of obtaining a response within the study’s timeframe.

## 3. Results

Twenty-eight articles fulfilled the inclusion criteria: clinical cases (*n* = 5) [15,16,17,18,19], case series (*n* = 3) [5,7,20], observational studies (*n* = 14) [12,13,23,24,26,27,28,29,30,31,32,35,38], and clinical trials (*n* = 6) [25,30,33,34,36,37]. Regarding its global distribution, 175 patients were found with a diagnosis of endonyx onychomycosis in Asia (*n* = 71), Africa (*n* = 60), America (*n* = 15), and Europe (*n* = 14) (Table 2).

In Asia, 71 cases have been reported to date, with India having the highest number (*n* = 56), followed by Iran (*n* = 9) and Taiwan (*n* = 3). Likewise, in Africa, a total number of 60 cases were found distributed in four countries: Botswana (*n* = 47), Cameroon (*n* = 9), Egypt (*n* = 2), and Somalia (*n* = 2). In America, 15 cases have been diagnosed, and the most affected countries have been the United States (*n* = 10), Brazil (*n* = 4), and Argentina (*n* = 1). Europe is the continent with the lowest number of diagnosed cases, with a total of 14, present mainly in Serbia (*n* = 6) and France/Italy (*n* = 3). In addition, 15 cases were registered in France, together with its overseas departments and territories (French Caribbean, French Guyana, and Reunion Island), located in America, Africa, and Europe. As they could not be traced to a single continent, they are presented independently (Table 2 and Figure 2).

Five clinical cases, three case series, and one non-randomized control trial [21] included a detailed description of the diagnosed patients. From them, a total of fifteen immunocompetent patients were registered—nine men and six women—with an age range from 7 to 69 years old. The characteristics described by Tosti et al. [5] in 1999 were used to guide us when determining the similarities and differences with the other published reports. The cases were divided for the analysis according to the etiological agent: six cases of *Trichophyton soudanense*, six of *T. rubrum*, one of *T. violaceum*, one of *T. tonsurans*, and one of *Fusarium* spp. The summary of the epidemiological, clinical, mycological, and histopathological characteristics from these cases is presented in Table 1.

Among the remaining studies, one randomized controlled trial [33] provided an explicit description; eight observational studies cited bibliography where endonyx onychomycosis was presented as milky white discoloration, without hyperkeratosis or onycholysis [39,40], accounting for 47.3% of the included articles. In contrast, six observational studies and three controlled trials did not explicitly reference the definition of endonyx onychomycosis, thereby lacking a traceable criterion for diagnosis in these cases. This group represented 47.3% of the reviewed studies. Additionally, one controlled trial (5.2%) defined it as a synonym of SWO [36], further indicating inconsistencies in terminology and diagnostic criteria across the literature (Table 3).

## 4. Discussion

The clinical, histopathological, and mycological features of endonyx onychomycosis were initially described by Tosti et al. [5] and later incorporated into Baran et al.’s new classification [6].

Following a systematic evaluation of the definitions employed in the analyzed papers, of the one hundred and seventy-five cases reported in 28 articles, along with the four cases originally described, seventy-seven cases were identified that referenced or detailed features consistent with the original description. Of these, four articles were based on Baran’s et al. [6] new classification, four cited other sources with a similar concepts [39,40], one study described a compatible definition, and eight described a clinical case compatible with the original report. In all 81 patients, clinical-mycological diagnosis was performed. In contrast, only 13 cases (16.04%) were assessed through a complementary histopathological analysis, in most cases using nail clipping.

To examine the traits outlined in the clinical cases, we chose to categorize them based on the etiological agent. *T. soudanense* and *T violaceum* were the first causative agents described; rarer pathogens are *T. rubrum* and *T. tonsurans*. For *T. soudanense*, there were three clinical reports; one is the original report by Tosti et al. [5]. When comparing the other three cases [15,20] to the previously mentioned description (Table 1), the milky white discoloration is present but characterized by patches rather than diffuse white discoloration. In the case of the two Somali siblings, the nail plate surface was altered by lamellar peeling, transverse indentations, coarse pitting, and, in the girl’s case, presented scaling over the hyponychium.

It is important to note that two articles were found during citation searching, which correspond to clinical descriptions prior to the introduction of the endonyx concept, as described by Tosti et al. [5]. These papers report cases with similar characteristics but caused by *T. violaceum* (Felix Sagher, 1948 [7], and James Hebert Graham, 1972 [14]). However, only the first one could be retrieved. Although it describes only one case of onychomycosis due to *T. violaceum*, it provides a key element for the histopathological definition of endonyx onychomycosis. Sagher observed that the hyphae were not causing any inflammatory reaction in the surrounding nail plate and noted that there were no fungi present in the subungual area, although the patient had a wavy and thick nail [7].

Mulvaney et al. reported features consistent with endonyx invasion but caused by *T. rubrum*, an agent not previously reported [16]. It was a case without hyperkeratosis, onycholysis, and with normal nail thickness. The nail surface was mostly normal except for distal lamellar splitting. Histopathology of a nail clipping revealed abundant hyphae invading the nail, except for the subungual keratin, and there was no inflammatory response. This point is crucial according to Tosti et al. [5] and Sagher [7]. *T. rubrum* could be another potential causative agent of endonyx onychomycosis, as has been seen in the cases reported by Souza et al. [21] and Maki et al. [17]. More histopathological evidence could provide stronger support for these fungi as etiological agents.

In the case of the patient diagnosed with fingernail endonyx onychomycosis caused by *T. tonsurans* [18], the authors did not report the necessary histopathological evidence to demonstrate the absence of fungal elements in the nail bed, nor any inflammatory changes around the fungi, although clinical characteristics were compatible with the original concept, and it is important to recall that the patient presented a concomitant *tinea corporis*, thereby suggesting a possible invasion mechanism.

Additionally, a case of *Fusarium* spp. was described by Negroni et al. [19]; the fungus was isolated independently in two samples that were taken 45 days apart—a result necessary for non-dermatophytic mold onychomycosis diagnosis [41,42,43,44]. However, they referred to onycholysis and a nail thickness greater than 3 mm. Furthermore, the histopathological analysis reported hyphae in the nail plate but did not mention whether there was an absence of inflammatory reaction around the fungus or in the nail bed.

This information highlights the need for further studies to determine whether other fungi could be potential etiological agents of this type of onychomycosis.

Overall, we identified 14 cases that exhibited all the characteristics of the endonyx onychomycosis definition. Interestingly, there seems to be a relationship between the patients’ age, the location of the affected nail, and the presence of other dermatophytic infections. In prepubescent patients, infection of the scalp or colonization by *T. soudanense* seems to be the route through which fingernails become infected, explaining why fingernail infections are more common in this group. In contrast, in adults, the infection tends to be more frequent on toenails and associated with tinea pedis, where *T. rubrum* is more common. This is consistent with the infection patterns and age distribution observed in other clinical types of onychomycoses [20,45]. Additionally, three different aspects of the milky white discoloration have been described: patches, longitudinal bands, and diffuse.

The infection mechanism of endonyx onychomycosis reaches the nail plate through the pulp of the fingers and invades the superficial and deep parts, thus causing a laminar separation of the nail without causing onycholysis. However, endonyx onychomycosis is characterized by damage to the inner surface of the nail plate, presenting a diffuse milky white discoloration in the infected nail that occurs due to the presence of tunnels filled with fungal elements in the thickness of the nail plate [5].

The analysis of the observational studies and control trials revealed that 52.5% of the studies exhibited a high risk of measurement bias, mainly due to unclear cited references, but also to the absence of detailed information regarding the clinical features of the reported cases. This limitation prevents an accurate evaluation of the variant-specific manifestations of endonyx onychomycosis. Furthermore, even among studies that cited or described a concept in alignment with the original definition, only one provided sufficient detail regarding the case descriptions. This lack of specificity has resulted in a significant gap in the literature, leading to missing or incomplete data spanning over two and a half decades of research. Moreover, most studies analyzed their data at the global level, describing little information to provide specific insights into the endonyx variant. Given these limitations, further research specifically focused on endonyx onychomycosis is necessary to enhance diagnostic precision and reduce the risk of bias in future studies.

The decision not to contact the authors of studies with limited case information due to time and resource constraints represented a limitation to the review processes used. This was because 67.8% of the data was missing and affected the comprehensiveness of the review process. Although, even if the authors had been contacted, it is unlikely that all missing information would have been available.

## 5. Conclusions

Onychomycosis endonyx is a rare variant that has a higher prevalence in the continents of Asia (40.5%) and Africa (34.2%), particularly in countries such as India (32%) and Botswana (26.8%). Nevertheless, a significant gap in knowledge was identified during the analysis, especially in etiology, treatment, and prognosis. The clinical and etiological diversity of the cases studied during this review highlights the complexity of their diagnosis and classification, with *Trichophyton soudanense* and *T. violaceum* being the most common etiological agents, although other fungi such as *Fusarium* spp., *T. rubrum*, and *T. tonsurans* have also been implicated in rare cases. Additional studies are required to clarify the clinical significance of each of these etiological agents.

The analysis of the clinical features typically presents with a milky white discoloration, without hyperkeratosis or onycholysis; however, there are discrepancies in the reported cases, such as the description of nail surface involvement, and their diagnostic methods, such as the lack of consistent histopathological evidence due to its performance not being mentioned in the studies, or due to differences in the clinical characteristics described, which suggest the need for further research to better understand the full range of manifestations and the responsible fungal agents.

With all the variability for endonyx onychomycosis found in the literature, analyses to draw conclusions on the prognosis or treatment possibilities are difficult. Therefore, emphasis on clinical description and histopathological analysis is essential for confirming the role of less commonly reported fungi. More accurate diagnostics following the points mentioned by Tosti et al. regarding clinical findings, sample collection, and pathological studies for this condition are mandatory.

Meanwhile, the most relevant lesson lies in the fact that the classical presentation of onychomycosis has nail thickness and surface alterations, but in exceptional cases, such as in endonyx onychomycosis, the nails do not present surface or thickness alterations and display only patches, longitudinal bands, or diffuse milky white discoloration. This rare clinical presentation differs significantly from the more common findings that typically guide the diagnosis of onychomycosis. Therefore, it is essential to consider this possibility in the approach to nail pathologies.

## Figures and Tables

**Figure 1 diseases-13-00110-f001:**
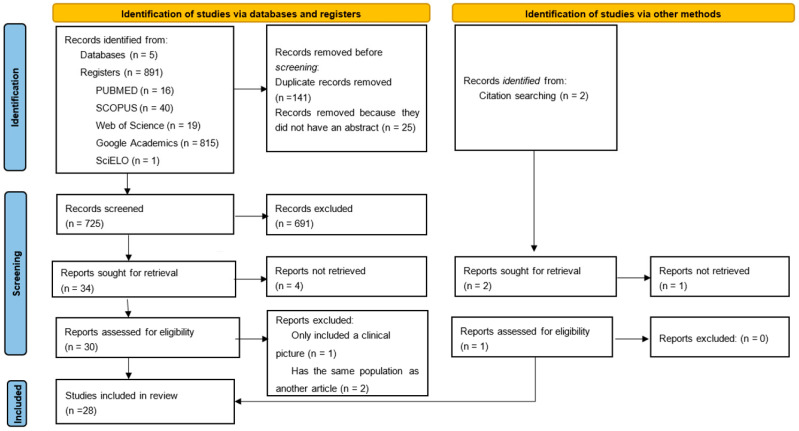
PRISMA 2020 flow diagram.

**Figure 2 diseases-13-00110-f002:**
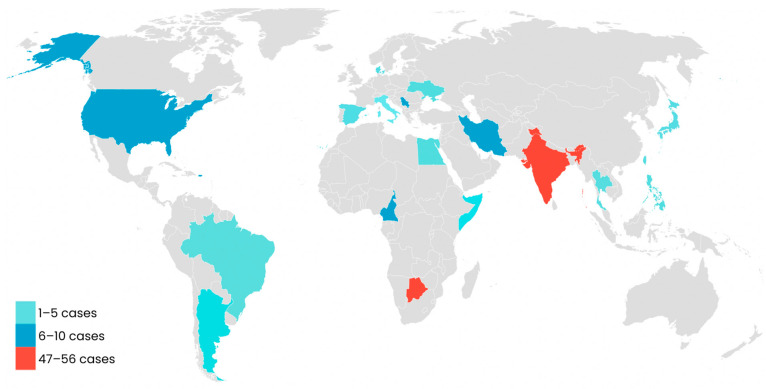
Global distribution of patients diagnosed with endonyx onychomycosis. Note: The 15 cases reported in France, together with its overseas departments and territories (French Caribbean, French Guyana, and Reunion Island), could not be traced to a single location, so they are not shown on the map.

**Table 1 diseases-13-00110-t001:** Clinical reports compared to Tosti et al. [5] description. Comparison between the originally described characteristics and the recent characteristics described in clinical reports and a clinical trial with diagnosed endonyx onychomycosis.

Epidemiology		Sex	M	F	F	M	M	M	M	F	M	F	M	F	M	M	F
		Age (in years)	65–69			19	40	45	51	25	61	9	1	43	65	39	28
		Nail localization	T	T	T	NA	T	T	Fn	Fn	T	Fn	Fn	T	T	T	T
		No. of affected nails	1			2	1		1		1	4	3	NA			
		Evolution (years)	NA			9	0.91	8	NA	0.42	3	1					
Clinical findings	Originally described	Absence of nail bed hyperkeratosis	X	X	X		X		X	X		X	X	X	X	X	X
		Absence of onycholysis	X	X	X		X	X	X	X		X	X				
		Normal nail thickness	X	X	X			X	X	X		X	X				
		Normal nail plate surface	X	X	X		X		X	X							
		Diffuse milky-white discoloration	X	X	X	X			X								
	Recently described	Transvers milky white band						X			X						
		Milky-white patches					X					X	X				
		Transvers indentations of nail plates				X				X		X	X	X	X	X	X
		Lamellar peeling								X		X	X				
		Coarse pitting										X	X				
		Scaling over the hyponychium										X					
		distal split												X	X	X	X
		Nail hyperkeratosis				X					X						
		Minimum subungual hyperkeratosis						X						X	X	X	X
		Onycholysis									X						
		Cubbing							X								
		Affection of the lunula						X			X			NA			
Sample collection	Originally described	1	X	X	X				X								
		2	X	X	X	X	X		X	X	X	X	X	X	X	X	X
Pathological study		3	X	X	X	X	NA	X	NA		X	NA					
		4	X	X	X	X											
		5	X	X	X			X									
		6	X	X	X	X		X									
		7	X	X	X												
		8	X	X	X			X									
		9	X	X	X												
Etiological Agent		*T. soudanense*					X					X	X				
		*T. violaceum*				X											
	Recently described	*T. rubrum*						X	X					X	X	X	X
		*T. Tonsurans*								X							
		*Fusarium* spp.									X						
Associated tineas		Tinea capitis				X						X	X				
		*T. soudanense*										X	X				
		Dermatophytid reaction											X				
		Tinea pedis	X	X	X												
		*T. soudanense*	X	X	X												
		Tinea corporis								X							
		*T. tonsurans*								X							
Reference			[5]			[7]	[15]	[16]	[17]	[18]	[19]	[20]		[21] !			

F—Female. M—Male. T—Toenail. Fn—Fingernail. NA—Not available. Characteristics originally described: 1. The nail plate was so firmly attached to the nail bed that clipping to obtain subungual debris produced pain and diffuse bleeding. 2. Direct microscopy with filaments. 3. Abundance of fungal elements in the nail plate. 4. Absence of fungal elements in the nail bed. 5. No hyperkeratosis. 6. Nail bed showed no inflammatory changes. 7. Normal hyponychium, without hyphae. 8. Normal hyponychium, without hyphae. 9. Fungi almost exclusively confined to the ventral nail plate. ! Clinical trial.

**Table 2 diseases-13-00110-t002:** Global distribution of reported cases with a diagnosis of endonyx onychomycosis.

Continent	Country/Region	Region	No. of Cases per Article	No. of Cases per Country	No. of Cases per Continent	References
Africa	Botswana	Gaborone	45	47	60	[22]
			2			[23]
	Cameroon	Buea and Limbe (Fako)	9	9		[24]
	Somalia	-	2	2		[20]
	Egypt	Portsaid	2	2		[25]
Asia	India	Muzaffarnagar	30	56	71	[26]
		Kolkata	15			[27]
		Delhi	5			[14]
		Mumbai	2			[28]
		Puducherry	2			[12]
		Shillong	1			[29]
		Chandigarh	1			[30]
	Iran	Sari	9	9		[31]
	Taiwan	Chiayi	3	3		[32]
	Thailand	Bangkok	1	1		[18]
	The Philippines	-	1	1		[15]
	Japan	-	1	1		[17]
America	United States of America	California	8	10	15	[33]
		New York	1			[7]
		Rhode Island	1			[16]
	Brazil	Montes Claros	4	4		[21]
	Argentina	Buenos Aires	1	1		[19]
Europe	Serbia	Sremska Mitrovica	6	6	14	[34]
	France/Italy	Cannes/Bologna	3	3		[5]
	Denmark	Roskilde	2	2		[35]
	Ukraine	Kiev	2	2		[36]
	Spain	Barcelona	1	1		[37]
America, Africa, Europe	France and Overseas Departments and Territories	France, French Caribbean, French Guiana, Réunion	15	15	15	[38]

**Table 3 diseases-13-00110-t003:** Assessment of the risk of bias through the analysis of the references cited to define “Endonyx onychomycosis” and the diagnostic approach performed.

No.	Study Design	Reference Cited					Diagnostic Approach Performed				Reference
		A New Classification of Onychomycosis [6]	Other	Written Concept	Unclear	Synonym of White Superficial Onychomycosis	Clinical Features	Direct Microscopy	Histopathology	Culture	
1	Observational study	X					X	X		X	[28]
2	Clinical Trial					X	X			X	[36]
3	Clinical Trial				X		X	X		X	[37]
4	Observational study		X, [40]				X	X		X	[27]
5	Observational study	X					X	X		X	[29]
6	Observational study				X		X	X		X	[22]
7	Observational study		X, [39]				X	X	X ^1^	X	[13]
8	Observational study	X					X	X		X	[26]
9	Observational study				X		X	X	X *	X	[12]
10	Observational study				X		X	X *			[38]
11	Clinical trial				X		X	X		X	[25]
12	Observational study				X		X	X		X	[31]
13	Clinical trial				X		X	X		X	[34]
14	Clinical trial			X			X			X	[33]
15	Observational study				X		X	X		X	[24]
16	Observational study		X, [39]				X	X		X	[22]
17	Observational study		X, [39]				X			X	[30]
18	Observational study				X		X	X		X	[23]
19	Observational study	X					X		X ^2^	X	[32]

Nail sample was taken through: ^1^ Nail punch or nail clipping; ^2^ Nail clipping. Note: * Not performed in all patients, so it is unclear whether endonyx onychomycosis-diagnosed patients were evaluated with this technique.

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
