# Peer review of "Onychomycosis Endonyx: A Systematic Review"

_diseases, 2025, doi:10.3390/diseases13040110_

Round 1

Reviewer 1 Report

Comments and Suggestions for Authors

The authors perform a literature review of studies citing the syndrome onychomycosis endonyx, a particular type of fungal nail infection, focusing on clinical and histopathologic description, diagnostic approach, and causative agents. They note marked variability across studies in these areas, including substantial divergence from the original description, either from specific differences or from lack of particular data collection. They conclude: “Emphasis in the clinical description and histopathological analysis are essential to confirm the role of less commonly reported fungi and more accurate diagnostic and treatment protocols for this condition are mandatory.”

The study is relatively quite comprehensive, casting a wide net and starting with many papers for consideration. The authors are also very detailed in their description of the winnowing of this collection, leading to final consideration of only a handful of papers. This is useful, but also emphasizes the relatively rarity of this condition, which may limit the impact of the study. Their conclusions about limitations and variability in the literature appear valid.

However, significance is limited. The authors may be correct, but what is the impact, in particular in terms of prognosis and treatment? The condition is relatively rare and may be challenging to diagnose. Fungal skin and nail infections are often diagnosed clinically from appearance and other evident features without the delay and expense of diagnostic approaches that may not influence disease course or treatment. The authors include the term “treatment” in arguing that specific diagnoses and identification of specific etiologic agents are important, which is theoretically self-evident, but do not provide support or cite evidence for this contention in the case of this syndrome, from their literature review. This aspect weakens the significance and justification of this study. Did the deficiencies or variability of diagnosis they found influence clinical course or treatment?

Author Response

Thank you for your time reviewing the manuscript. Your comments and suggestions improved its quality.

You will find the responses to your comments attached.

Reviewer 2 Report

Comments and Suggestions for Authors

Please find attached

Comments on the Quality of English Language

There are few grammar and sentence formations that could be improved.

Author Response

(The authors gave the same response as above.)
